# Theoretical principles of transcription factor traffic on folded chromatin

Ruggero Cortini [1,2] & Guillaume J. Filion [1,2]

All organisms regulate transcription of their genes. To understand this process, a complete understanding of how transcription factors find their targets in cellular nuclei is essential. The DNA sequence and other variables are known to influence this binding, but the distribution of transcription factor binding patterns remains mostly unexplained in metazoan genomes. Here, we investigate the role of chromosome conformation in the trajectories of transcription factors. Using molecular dynamics simulations, we uncover the principles of their diffusion on chromatin. Chromosome contacts play a conflicting role: at low density they enhance transcription factor traffic, but at high density they lower it by volume exclusion. Consistently, we observe that in human cells, highly occupied targets, where protein binding is promiscuous, are found at sites engaged in chromosome loops within uncompacted chromatin. In summary, we provide a framework for understanding the search trajectories of transcription factors, highlighting the key contribution of genome conformation.

[1] Genome Architecture, Gene Regulation, Stem Cells and Cancer Programme, Centre for Genomic Regulation (CRG), The Barcelona Institute of Science and Technology, Dr. Aiguader 88, 08003 Barcelona, Spain. [2] Universidad Pompeu Fabra (UPF), 08003 Barcelona, Spain. Correspondence and requests for materials should be addressed to R.C. (email: ruggero.cortini@crg.eu) or to G.J.F. (email: guillaume.filion@gmail.com)

Transcription factors play a key role in the regulation of transcription[1]. Upon binding to cognate regulatory sequences within the chromatin, the transcription factors trigger a cascade of molecular events culminating in the recruitment of the RNA polymerase and subsequent transcription[2]. Understanding how transcription factors find their targets in the genome is thus the very first and pivotal step in understanding how transcription is controlled at the molecular level.

Transcription factors contain a DNA-binding domain with a non-specific affinity for DNA and a high specific affinity for some target sequence called the binding motif[3]. Affinity for their motifs predicts the binding of transcription factors in vitro[4]. However, recently acquired data using the ChIP-on-chip and ChIP-seq technologies showed a different picture in the nucleus of multicellular eukaryotes. Transcription factors occupy sites where their motif is absent[5,6], and leave unbound most sites where it is present[7–10], indicating that a simple protein–DNA interaction model is insufficient to describe the transcription factor dynamics in vivo.

The nucleus is a rich and heterogeneous environment, where higher-order interactions between molecules take place. Histones, the core component of chromatin itself, present a barrier to transcription factors binding. To overcome this constraint, the so-called pioneer transcription factors have been shown to bind their targets in the presence of a nucleosome, or even aid other transcription factors to gain access to their targets[11]. However, as the binding of pioneer transcription factors is vastly different between cell types[12], the simple view that they bind their motif regardless of the chromatin context cannot hold true. Therefore, additional phenomena dictate and drive the search process of transcription factors within the nucleus.

One such mechanism is facilitated diffusion[13,14], which emphasizes the key role of non-specific affinity for the search kinetics. Transcription factors are first adsorbed onto DNA or chromatin through an electrostatic pull acting at a short distance[15], and for a short period of time they diffuse on the polymer. This scanning or sliding mode is essential to discover the target sequence[13,16]. However, both terms are somewhat inaccurate: transcription factors may merely detach and reattach to the chromatin immediately. At the points where the chromatin fibers meet, transcription factors can in theory fall off one fiber and reattach to the adjacent one, effectively jumping over large genomic distances while diffusing on the chromatin. Such effects have been observed in vitro, where the three-dimensional (3D) conformation of naked DNA was shown to impact the search kinetics[17].

The importance of this mechanism of diffusion is well established in the eukaryotic nucleus[18], and it is now clear that transcription factors diffuse intermittently in the nucleoplasm and on the chromatin fiber itself[19,20]. However, little is known about the impact of the geometry of the chromatin fiber. Genomes have a characteristic 3D structure, revealed by chromosome capture methods such as Hi-C[21]. It is still unclear how genomes acquire a particular conformation, however, it is known that once preferential contacts are established, they can influence the trajectories of transcription factors diffusing along the chromatin.

Therefore, it is tempting to hypothesize that the knowledge of chromosome conformation could be used to obtain an insight into the search dynamics of transcription factors. However, the general principles are presently unknown for lack of a general theory. Previous work by ourselves and others suggested methods to infer the binding profiles of proteins on DNA from Hi-C matrices[22–24], but the validity of those approaches remains unclear as there is no guarantee that they correspond to realistic physical processes.

Here we establish the basic principles of transcription factor diffusion on folded chromatin. We use a molecular dynamics simulation approach to investigate the role of chromosome conformation in the search process. Exploring configurations with the strings and binders model of Barbieri et al.[25], we find that geometry has a significant influence on the traffic of diffusing bodies with an affinity for the polymer. Strikingly, polymer loops increase traffic and occupancy in a wide range of conditions, but decrease them at high compaction. Consistently, Hi-C and ChIP-seq data show that massive protein binding accumulates at genomic loci engaged in long-distance contacts. Overall, this work suggests that the 3D conformation of the genome affects the discoverability of binding sites and contributes to the global distribution of transcription factors in vivo.

## Results

**Model and assumptions**. We used molecular dynamics simulations to understand how the conformation of the genome can affect diffusion of transcription factors. This modeling strategy captures the behavior of simple objects according to realistic physical interactions. To develop a general and tractable model, we stripped down the specific features of chromatin and transcription factors to their bare essentials: chromatin was considered as a folded polymer and transcription factors as diffusible molecules with an affinity for the chromatin.

We simulated spherical particles, referred to as tracers, which were used as a model for transcription factors except for one important difference: tracers have no specific affinity for any particular site on the polymer, contrary to the specificity of transcription factors. This aspect of the model is essential to capture the dynamics of transcription factors in search mode, rather than in bound mode. Using these restrictions, tracers can be considered as either non-specific transcription factors, or transcription factors not bound to their target site. The strength of the non-specific affinity of the tracer on the polymer was labeled $\varepsilon$.

Folded polymers were simulated using a model originally developed by Barbieri et al.[25] and are summarized graphically in Fig. 1. Briefly, the model describes the large-scale structure of the polymer as an aggregate of stable loops formed between predefined anchor monomers. The loops are formed by special particles called binders that have a high affinity for anchor monomers and can thus bridge them together. The overall openness or compaction of the polymer depends on the number of loops, or more accurately on the fraction of these anchor monomers, labeled $\phi$.

Varying $\varepsilon$ and $\phi$ allowed us to explore a wide range of conditions. For each simulation, we computed the intra-polymer contact matrix where each entry was the number of times two given monomers were in contact during the simulation (meaning that their distance is less than a threshold $t$, see Methods). From this matrix, we computed the row sum, referred to as the polymer contacts, representing the total amount of contacts for each monomer. We also computed the total number of contacts between the tracers and the monomers, referred to as the traffic of the tracers. This profile represents the total amount of time tracers spent in contact with different regions of the polymer.

Most of the explored parameter space corresponds to physiologically relevant conditions. For example, recent estimates from electron microscopy suggest that within the nucleus of a typical human cell, the local chromatin density varies in the range of 12–50%[26], which corresponds to values of $\phi$ in the range of 0.1–0.4 (see Methods and Supplementary Fig. 2a). Other experiments suggest that for a typical human cell, a transcription factor spends approximately the same amount of time diffusing on the chromatin and in the nucleoplasm[19,20]. This corresponds to $\varepsilon \approx 2k_BT$ in the simulations, and for $\varepsilon$ in the range of 1.0–2.5

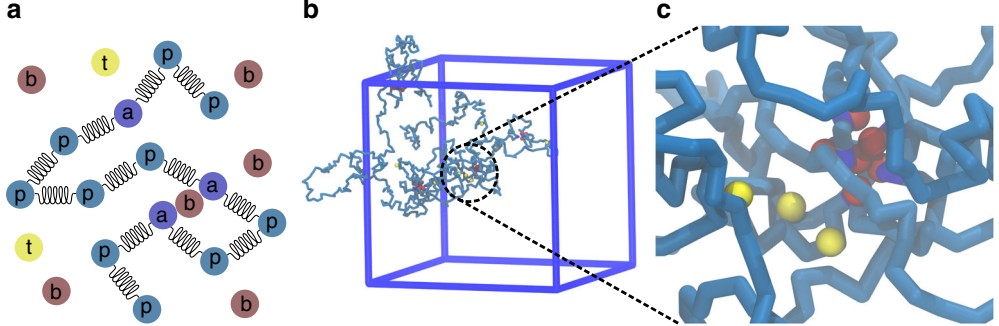

**Fig. 1** Description of the simulation setup. **a** The polymer is made of p (polymer) and a (anchor) particles, connected by harmonic springs. Binders (b particles) are introduced to bridge a particles. Tracers (t particles) interact non-specifically with both p and a particles. Binders and tracers interact only by hard-core repulsion, i.e., by volume exclusion. **b** Snapshot of a simulation together with the simulation box. **c** Zoom-in of a loop formed by the binders and their binding sites, along with the tracers bound nearby (yellow)

$k_BT$, the time spent on the polymer varies from 5 to 95% approximately, which covers most possible cases (Supplementary Fig. 3c).

We carried out two independent sets of simulations, the first with 10 tracers and the second with 200 tracers. This allowed us to test the effect of the tracers themselves on the conformation of the polymer. The results of both simulations were similar. However, it is clear from the analysis of both simulations that when the tracer affinity exceeds a given threshold, the polymer changes conformation due to a local increase in the tracer concentration (Supplementary Note 1). The implications of this result are discussed in more detail within the Discussion section.

Importantly, the results clearly demonstrate the robustness of the model in regard to the details of the simulation. In particular, simulations with monovalent tracers (Supplementary Note 4), and in a crowded medium (Supplementary Note 6), gave the same qualitative results.

**Polymer loops have two opposite effects on the tracer traffic**. In what follows, the tracer traffic is defined as the total number of times that the tracers are detected in contact with each monomer. In other words, tracers visit more often a monomer with high traffic than a monomer with low traffic. In our simulation assay, polymer looping has two opposite effects on the traffic of the tracers, which is more clearly demonstrated when the values of the parameters are extreme (Fig. 2).

Of note is the correspondence between the monomer contact frequency and the traffic of the tracers. The correspondence is particularly clear in the cases where only a few loops are present in the polymer (low $\phi$) and the binding of the tracers is strong (high $\varepsilon$).

We measured the correspondence between the polymer contacts and the tracer traffic using the Pearson correlation coefficient and the Kullback–Leibler (KL) divergence $D_{KL}$. Here, the KL divergence is computed as:

$$D_{KL}(C|R) = \sum_i C_i \log \frac{R_i}{C_i}, \qquad (1)$$

where $C_i$ and $R_i$ represent the traffic on the $i$th monomer and the polymer contacts made by monomer $i$, respectively. The KL divergence and the Pearson correlation coefficient are both valid metrics although they represent different features of the data. When using the Pearson correlation coefficient, the two variables are interpreted as numeric random variables and the quantity of interest is their covariation. When using the KL divergence, the two variables are interpreted as probability distributions and the

quantity of interest lies in the information lost by using the polymer contacts array instead of the tracer traffic array.

If the polymer has few loops ($\phi = 0.02$) the high values of $r$ and the low values of $D_{KL}$ indicate that the total amount of contacts coincides with the traffic of the tracers (Fig. 2b). This is a direct result of the combination of the two following effects: Firstly, the on-rate of the association reaction is doubled at the contact points because tracers come from two distinct branches of the polymer. Secondly, the off-rate reduces as the local concentration of the polymer increases, thereby favouring binding (see Supplementary Movie 1). The latter effect is due to the attractive component of the interaction potential between the tracers and the monomers. The overall effect is an increase of the ratio between on-rate and off-rate, i.e., a net increase in traffic.

Highly compacted polymers with a multitude of looping sites ($\phi = 0.50$) define another category. High compaction combined with low tracer–polymer affinity (low $\varepsilon$) results in a strong anti-correlation between the contact amount and the occupancy profile of the tracers, i.e., the tracers have a tendency to visit the sites that make fewer contacts. The reason is that the polymer forms a globule, where the most visited monomers are at the surface, and their contacts with other monomers are less frequent. In comparison, when the tracer–polymer affinity is high (high $\varepsilon$) we observe a caging effect, whereby the tracers remain blocked in compacted inner structures (see below). In this case, both $r$ and $D_{KL}$ report that the tracer traffic and the polymer contacts are unrelated (see Supplementary Movie 2).

Figure 3 summarizes these results with the average values of the KL divergence and the Pearson correlation coefficient for all the parameters tested in our simulations. Regardless the number of tracers, a larger number of loops results in a poorer correspondence between the traffic of the tracers and the contacts of the polymer. In summary, the conformation of the polymer defines two regimes: one where the polymer is uncompacted and contacts predict the traffic of the tracers, and another one where the polymer is compacted and contacts do not predict the traffic of the tracers.

**High polymer compaction excludes tracers**. Why does the conformation of the polymer cease to predict the traffic of the tracers when the number of loops increases? At least two non-mutually exclusive scenarios can be imagined. In the first, the polymer forms globular domains that are too dense for tracers to enter. In the second, tracers enter the domains but steric effects prevent them from binding to the anchors of the loops.

To address this question, we defined the coverage of the polymer as the percentage of monomers visited by a tracer at least

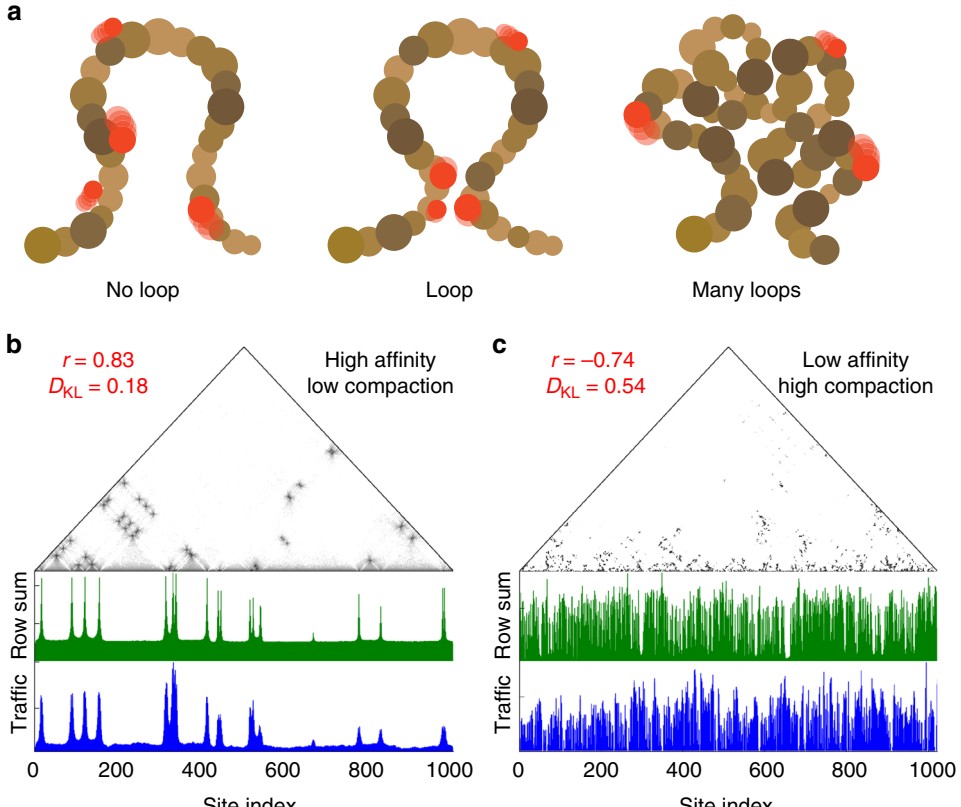

**Fig. 2** The dual role of looping on transcription factor traffic. **a** Effect of chromosome conformation on the traffic of transcription factors. At low compaction (left), transcription factors (red) slide uniformly on the chromatin fiber (brown). In the presence of a loop (middle), transcription factors accumulate at the contact point. When the number of loops is high (right), transcription factors diffuse on the outer shell of the globule due to volume exclusion effects. **b**, **c** Examples of simulation results with the smallest and largest values of $\phi$ and $\varepsilon$ in cross combinations. For each example, we show the contact matrix of the polymer in log scale, above the traffic of the tracers and the total amount of contacts for each monomer (Row sum). Each plot indicates the values of the Pearson correlation coefficient $r$ between the two, along with their Kullback–Leibler divergence (see text). Panel **b** shows a simulation with low polymer compaction and high tracer affinity, resulting in a strong correlation between the polymer contacts and the tracer traffic. Panel **c** shows that the opposite happens with a low tracer affinity and high polymer compaction

once during the simulation. Figure 4 shows that the coverage decreases as the number of loops increases (high $\phi$), consistent with the view that the tracers cannot access the core of the polymer. However, increasing the affinity of the tracers alleviates this effect, so the exclusion does not proceed by hard-core repulsion, otherwise even tracers with high affinity for the polymer could not penetrate the core. In fact, most of the monomers are accessible at some value of the non-specific affinity, so for the highest values of the compaction that we tested, the polymer rarely has a density as to be completely impermeable to tracers. We surmise that higher values of the compaction $\phi$ would result in a fully impenetrable polymer core.

We computed the average correlation between the occupancy of the tracers and the occupancy of the binders. The binders bridge the loop anchors and remain fixed on the same monomers for the duration of the simulation. As the number of loops increases (high $\phi$), the correlation of the occupancy profiles becomes strongly negative, indicating that the binders exclude the tracers at looping sites. This can be partially alleviated by a higher non-specific affinity; however, the correlation remains negative at high number of loops.

These results suggest that in compact polymers, both effects apply: the core of the polymer domains becomes less accessible and the anchor sites become crowded. As a result, contacts within the polymer become poor predictors of the occupancy of the tracers.

To better understand the contribution of volume exclusion, we tested the behavior of tracers interacting with the polymer only through hard-core repulsion. We kept the polymer in a fixed state while simulating the motion of the tracers alone. In terms of traffic, a fixed polymer gives essentially the same results as a mobile polymer (since the simulated polymer structures are quite rigid) but substantially speeds up the simulations when modeling tracers without affinity for the polymer.

We ran the simulations at two extremes of the compaction spectrum (Fig. 5). At low polymer compaction ($\phi = 0.02$), tracers with and without affinity for the polymer have dissimilar traffic profiles ($r = 0.02$). This confirms the importance of non-specific affinity in shaping the traffic. In contrast, at high polymer compaction ($\phi = 0.50$), tracers with and without affinity for the polymer have comparable traffic profiles ($r = 0.58$). This shows that in this regime, the affinity between the polymer and the tracers has little impact on the occupancy. In conclusion, the pattern of tracer traffic in highly compacted polymers is mainly driven by volume exclusion effects.

**Highly occupied targets in the human genome.** An important question is whether the phenomena observed in our simulations are relevant to living cells. There is a general agreement that chromatin compaction tends to exclude transcription factors[27], but the idea that the geometry of the chromatin may guide transcription factors is still exploratory[22,24,28,29].

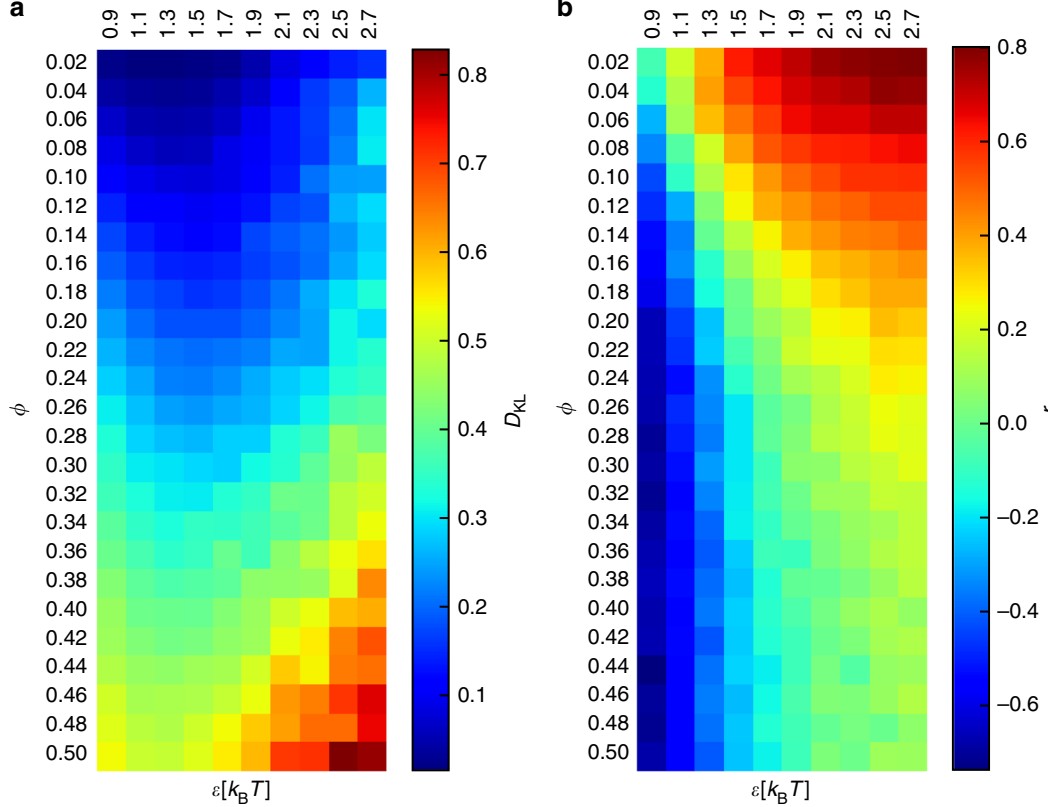

**Fig. 3** Coincidence of the traffic of the tracers with the contacts of the polymers. Average values of the Kullback–Leibler divergence (**a**) and of the Pearson correlation coefficient (**b**) for different model parameters. Calculations were performed as described in Methods

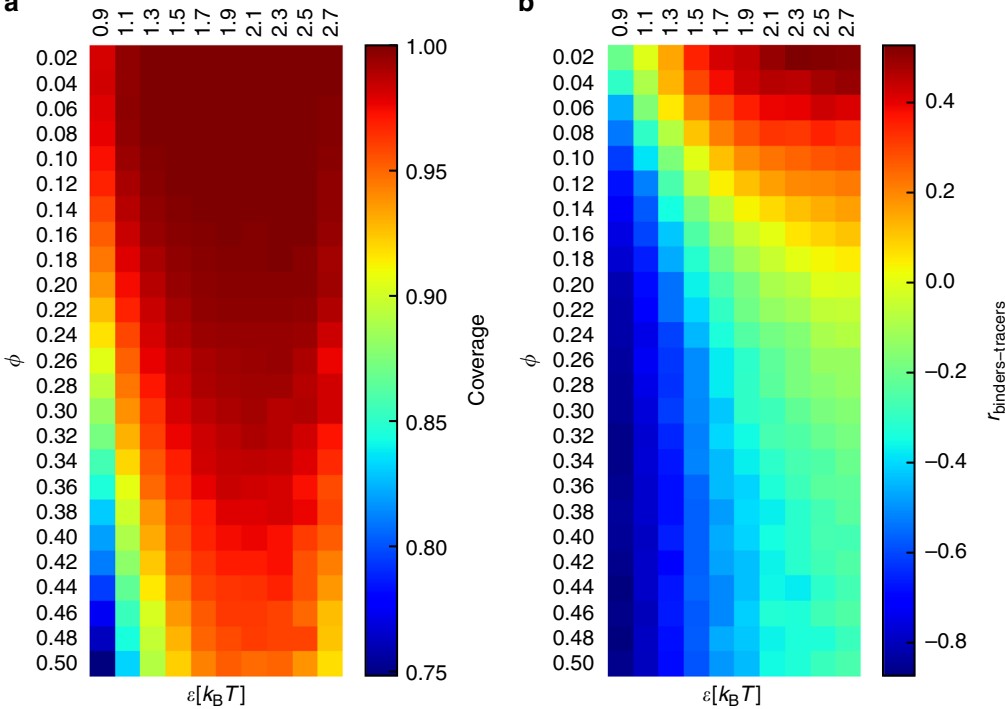

**Fig. 4** Volume exclusion effects in the simulations. **a** Average percentage of sites of the polymer that are visited by the tracers during the simulations (coverage). **b** Average values of the Pearson correlation between the tracer and binder occupancy sets

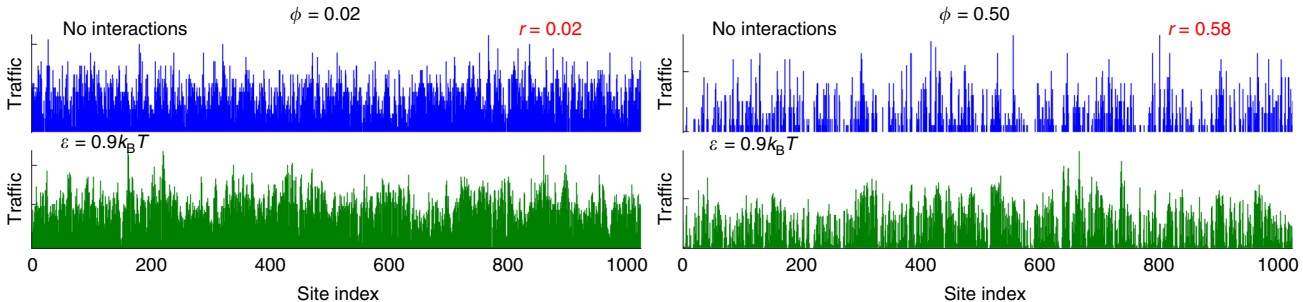

**Fig. 5** Traffic with and without tracer–polymer interactions. In blue: traffic of tracers without tracer–polymer interactions, i.e., interacting only through hard-core repulsion. In green: traffic of tracers with tracer–polymer affinity equal to $0.9k_BT$. Left: low compaction with $\phi = 0.02$. Right: high compaction with $\phi = 0.50$. Values of the Pearson correlation coefficient between the polymer occupancy and the tracer traffic are reported in red

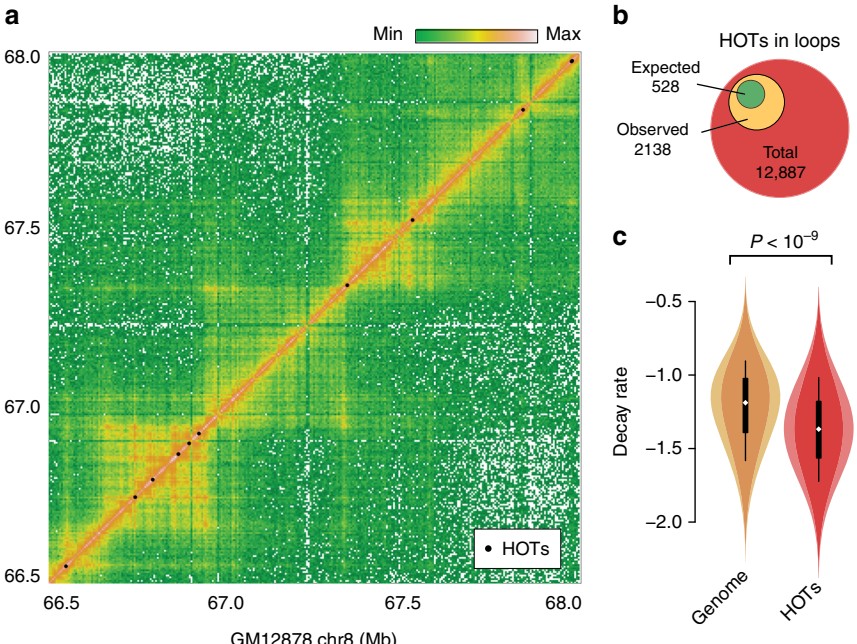

**Fig. 6** Highly occupied targets in the human genome. **a** Hi-C map of the human cell line GM12878, along with the position of the highly occupied targets (HOTs) indicated by black dots. **b** HOTs are enriched at chromosomal loops in GM12878. The location of the HOTs from Foley and Sidow[30] was crossed with the location of the loops from Rao et al.[30,31]. Out of 12,887 HOTs, 2138 were found in loops versus 528 expected (binomial distribution $P < 10^{-9}$). **c** HOTs are located in uncompacted chromatin. The violin plots show the distribution of contact decay in 50 kb windows in the whole genome versus each HOT. The mean values are −1.22 and −1.37, respectively (95% confidence interval for the difference 0.146–0.157, Student's $t$ test with Welch–Satterthwaite correction and 18,080 degrees of freedom $P < 10^{-9}$). The white dot inside each violin represents the median of the distribution. The black block ranges from the first to the third quartile and the whiskers range from the first to the ninth decile

Our results indicate that at low chromatin compaction, long-range chromosomal contacts may increase the traffic of transcription factors. The simulations predict a 2–5-fold increase of non-specific binding (Fig. 2). The effect is modest, but it applies to every molecule with an affinity for chromatin and with a size comparable to that of the polymer (see Supplementary Note 5).

It is possible that loop-enhanced traffic may explain one of the least understood patterns of transcription factor occupancy, namely the existence of highly occupied targets[5,6,30] (HOTs). HOTs, or binding hot spots, are small regions of the genome (typically less than 1 kb) bound by most proteins with an affinity for chromatin. The promiscuous binding of transcription factors in HOTs is independent of the presence of their binding motifs, suggesting that the process depends on non-specific affinity and protein–protein interactions. Could HOTs be a result of the formation of chromatin loops?

To test this hypothesis, we used high-resolution Hi-C data performed in the human cell line GM12878[31], and HOTs locations obtained from the ChIP-seq profiles of 96 proteins mapped in the same cell line[30]. It is visually apparent that HOTs tend to localize at regions engaged in long-distance chromosomal contacts (Fig. 6a). HOTs are strongly enriched at loop anchors, as defined by Rao et al.[30,31]. Indeed, 2138 of the 12,887 HOTs lie at the basis of a loop, compared to 528 expected (Fig. 6b). Thus the trend is very robust, especially considering the imprecision in calling loops and HOTs.

The simulation results predict that HOTs should be present in the regions of the genome that are the least compact. The standard way to compare compaction levels is to estimate the local rate of contact decay, which is a measure of the polymer state[25]. The probability of Hi-C contacts is locally proportional to $s^{-\alpha}$, where $s$ is the linear separation between the loci and $\alpha$ is the decay rate. The most compact states of the polymer correspond to

the lowest local values of $\alpha$ and vice versa. The local contact decay surrounding HOTs in the dataset is significantly lower than the genome average (Fig. 6c), indicating that HOTs tend to occur within the least compacted regions of the genome, consistent with the simulation results.

Overall, these results indicate that massive protein binding tends to accumulate at loop anchors in open regions of the nucleus, consistent with our prediction that chromatin loops enhance the traffic of transcription factors in the nucleus.

## Discussion

In this study, we used molecular dynamics simulations to establish the governing principles of transcription factor diffusion on folded chromatin (Fig. 2a). Despite the fact that our assumptions are reasonable, our model of chromosomes and transcription factors is simplistic, so we do not claim that it represents the real molecular detail at work in a cell nucleus. It only captures some general properties of the system.

The main feature of the polymer structure is the amount of contacts, acting in two opposite ways (Fig. 3): at low density, it increases the traffic of the tracers; at high density, the compaction of the polymer results in a volume exclusion effect that keeps the tracers away from loop anchors (Fig. 4). In the latter case, the tracers tend to interact with the monomers on the outside of globular domains, but they can also enter the structure, especially if their affinity for the polymer is high. These results are in line with experimental data obtained in live cells[27], where it was shown that chromatin compaction excludes diffusible factors, but no compartment is fully inaccessible.

Importantly, these effects describe the behavior of transcription factors in search mode only. Theoretical analyses[32] show that chromatin loops may increase the on-rate by a factor of 2–5 (the number of fibers at the contact point, see Fig. 2), but the off-rate may vary by several orders of magnitude with the DNA sequence[4]. A 2-fold effect is large for the binding kinetics, but negligible for the average occupancy of high affinity binding sites. Consistently with the view that chromosome conformation has a mild impact on the occupancy of high affinity binding sites, it was recently shown that target sites of many transcription factors remain bound on the mitotic chromosome[33], where the conformation is radically different[34].

Nevertheless, transcription factors spend a large fraction of time in search mode[19,20] and accumulate at HOTs, where most binding motifs are absent[6]. Our results suggest that this alternate binding mode depends on the conformation of the genome. The HOTs are found primarily at loop anchors in uncompacted regions (Fig. 6), as predicted by the simulations (Figs. 2 and 3). Importantly, these chromosome conformation effects apply to all molecules with an affinity for chromatin within the nucleus, provided the size of the particles is comparable to the pores of the chromosome folds (see Supplementary Note 5). This permits the appearance of emergent behaviors, induced by the interactions between the components.

Our model provides significant additional evidence to understand the formation of the 3D genome structure. In the molecular dynamics simulations, the structure of the polymer is given by the fixed distribution of the binders. This model recapitulates the essential features of Hi-C maps[25], but it does not correspond to the contentious issue of how the genome is actually folded[35]. For example, the loop extrusion model proposes the existence of molecular machines capable of creating loops (i.e., cohesins[36,37]). Other reports focused on the crucial role of epigenetic domains in shaping the 3D genome[38]. Furthermore, architectural proteins such as CTCF also play a crucial role in shaping the 3D genome[39], and they too may act as the binders in our simulations.

Nevertheless, the molecular details underlying the formation of the chromatin loop is not relevant to the present work. All that matters is that the loops are stable enough for the effect described above to take place.

We showed, as mentioned in the Results section, that when the number of tracers and their affinity exceed a critical value, the tracers themselves create intra-polymer contacts (Supplementary Note 2). This effect was predicted by the strings and binders switch (SBS) model of Barbieri et al.[25] and the bridging-induced attraction mechanism[40,41], in which loops are created in a non-specific fashion. Here, we show that it is also valid when the polymer has specific loops in its structure, due to the fact that we separated the role of specific and non-specific bridging particles. This indicates that a high concentration of transcription factors may by itself induce the chromatin fiber to collapse locally, a result which suggests that the many transcription factors present in HOTs may stabilize long-range contacts. More work is necessary to clarify the effect of heterogeneous binding profiles of the transcription factors in mediating and stabilizing chromatin loops.

A strong assumption will need to be tested: transcription factors are close to saturating concentration in the nucleus, so that the weak effect of a chromatin loop is sufficient to nucleate an aggregate. It will also be important to elucidate whether such aggregates are stable or transient, and what other factors are necessary for their formation.

It was previously observed that cohesin binding is frequently associated with the presence of many transcription factors[42], but the mechanism remained unclear. Since cohesin is necessary for the formation of the loops[43], we propose the following interpretation: cohesin binding induces the formation of a chromatin loop, which increases the local concentration of transcription factors. This nucleates the formation of an aggregate stabilized by protein–protein interactions. Finally, the high concentration of transcription factors compacts the chromatin the same way we observed for a large number of tracers, which further stabilizes the loop. Although we did not simulate cohesin loop formation explicitly, the binders in our simulations can be thought of as cohesin proteins. The time scale involved in the formation of cohesin loops is in fact very long[44], so that in our simulations we do not need to consider that the loops are dynamic. Once again, our model does not represent actual molecular mechanisms but only their essential properties regarding the diffusion of transcription factors.

This interpretation suggests that long-distance contacts may not be an accidental property of enhancers, but rather their essential mechanism of action. In this regard, the 3D organization of the genome could be a way to guide proteins of the same complex to the same locations in order to facilitate their assembly. This would also imply that the formation of liquid-like aggregates is dependent on the geometry of the chromatin polymer.

Our model is consistent with the experimental data regarding the distribution of HOTs in the nucleus. It also makes other predictions that can be experimentally tested. The first is that binding kinetics are 2–3 times faster when the target lies at the basis of a chromatin loop. Since more search trajectories lead to the target site when it lies within a loop, a rough estimate is that the search time is reduced by the number of chromatin fibers that meet at the cross point. The second prediction is that HOTs should disappear in mitosis. Indeed, the compaction of the chromosome and the disappearance of long-distance contacts and loops[34] are unfavorable conditions for the accumulation of transcription factors. Future experimental data will allow these predictions to be tested.

In summary, this study shows that the conformation of the genome should be taken into account to fully understand the

distribution of transcription factors in the nucleus of animal cells. More generally, the principles highlighted here pave the way for a general theory of transcription factor-facilitated diffusion on folded chromatin.

## Methods

**Simulation setup.** Our simulation model is the same as the SBS model[25], and consists of the following elements (see Fig. 1):

- A bead–spring polymer, consisting of $N$ total particles, which represent the individual monomers. The polymer is made of two types of particles: particles of type p (polymer), which have no special property, and type a, which are binding sites (anchors). The fraction of a particles in the polymer is the parameter $\phi$.
- Binders, type b, which are free to diffuse in space, have strong attractive interactions with the a particles on the polymer. In our system, we keep the number of binders such that there are always two binders per binding site. Therefore, the number of binders is $n_b = 2\phi N$. The interaction strength is chosen in such a way that the binders are ordered[45].
- Tracers, labeled t, diffuse freely in space and have an attractive short-ranged interaction to the entire polymer, that is, with both p- and a-type particles. The number of tracers is $n_t$.

The polymer is described as a classical bead–spring system. Each particle in our system has a radius $\sigma$. We chose $\sigma = 15$ nm independently of the particle identity, and $\sigma$ is set to be the length scale in the simulation units. The choice of the length scale is to match the approximate width of the chromatin fiber (30 nm fiber). Successive beads in the polymer are connected with harmonic springs of stiffness $k = 330 k_B T/\sigma^2$, as illustrated in Fig. 1. The interaction between successive monomers is then given by

$$E_{\text{monomer}} = \frac{1}{2} k \left( |\mathbf{r}_{i+1} - \mathbf{r}_i| - r_0 \right)^2, \qquad (2)$$

where $\mathbf{r}_i$ is the position vector of the $i$-th monomer, and $r_0 = 1.2\sigma$ is the rest position of the spring. Note that we set $k_B T = 1$ in the local simulation units.

All particles interact via the Lennard–Jones (LJ) potential with the other particles. The general interaction form is given by

$$E_{ij}(r) = 4\varepsilon_{ij} \left[ \left( \frac{\sigma}{r} \right)^{12} - \xi_{ij} \left( \frac{\sigma}{r} \right)^6 \right]. \qquad (3)$$

Here, the coefficients $\varepsilon_{ij}$ and $\xi_{ij}$ depend only on the particle identity (p, a, b, or t); $r$ is the center-to-center distance between the two particles. The parameter $\xi_{ij}$ is set to zero for particles that interact only through hard-core repulsion, and it is set to 1 for particles that have an attractive component. As usual, we apply a cutoff distance for the non-bonded interactions, set to $r_{\text{cut}} = 3\sigma$.

The binders were assigned a fixed, strong interaction energy with the binding sites, $\varepsilon_{ab} = 10 k_B T$. The interaction between the tracers and the polymer is given by $\varepsilon \equiv \varepsilon_{tp} \equiv \varepsilon_{ta}$. This is the other parameter that we vary in our simulations.

We used 25 different values of $\phi$, from 2 to 50%, in steps of 2%, and 10 different values of $\varepsilon$, from $0.9 k_B T$ to $2.7 k_B T$, in steps of $0.2 k_B T$. We simulated the system for all possible pairs of $\phi$ and $\varepsilon$. For each pair, ten independent simulations were carried out, each one with a different distribution of randomly placed binding sites (a sites) on the polymer.

The values of $\phi$ and $\varepsilon$ that we chose are such that a large portion of the explored parameter space corresponds to physiologically relevant conditions. Measuring the radius of gyration $r_{\text{gyr}}$ of the polymer (see Supplementary Note 2) allows us to probe the local density of the polymer by calculating

$$\rho_{\text{local}} = N \left( \frac{\sigma}{r_{\text{gyr}}} \right)^3. \qquad (4)$$

Experiments by Ou et al.[26] on human cells suggest that the local chromatin density varies approximately 4-fold, and that the highest observed local density is of the order of 50%. In our simulations, this corresponds to $\phi \approx 10–40\%$ (see Supplementary Fig. 2a). On the other hand, the experimental work by Normanno et al.[20] showed that the ectopically expressed TetR repressor spends about half of the time diffusing freely in the nucleoplasm and half of the time non-specifically bound to chromatin in human cells. As we show in Supplementary Fig. 3c, there is an excellent correspondence between the percentage of time that the tracers spend bound to the polymer and the value of $\varepsilon$. Fixing this value at 50% as in the case of TetR, we find $\varepsilon \approx 2 k_B T$, which is of the order of the weak hydrogen bonds formed by proteins transiently bound to chromatin. As $\varepsilon$ varies from $0.9 k_B T$ to $2.7 k_B T$, the time spent bound to the polymer ranges from approximately 5 to 95%, covering most possible values.

Molecular Dynamics (MD) simulations were carried out using the HOOMD-blue software[46,47]. To keep the temperature of the system constant at the value $T$,

the system was simulated using Langevin dynamics. To each particle of the system was assigned a drag coefficient $\gamma = 1$ (in local simulation units), equal for each particle. For free particles, this gives rise to a bulk diffusion coefficient $D_0 \approx 1.2$ cm$^2$/s in real units (see also Supplementary Note 3). We chose the simulation box to be cubic, with edge length $L = 50\sigma$. Periodic boundary conditions were applied in the simulations.

An equilibration of $2 \times 10^7$ time steps was run before taking samples of the system. After that, snapshots of the system were taken every $10^4$ steps. The total length of the MD runs was $10^8$ steps.

**Polymer contact matrix and tracer traffic.** To assess the relationship between the 3D structure of the polymer and the diffusing properties of the system, we evaluated the contact matrix of the polymer with itself. To obtain a result similar to chromosome capture-based interaction matrices, we evaluated a boolean contact matrix, which gives for each pair of particles a value of 1 if the two particles are in contact, and 0 otherwise. The criterion to establish whether the two particles are in contact is by assessing whether the center-to-center distance between the two particles is smaller than a given threshold, $t$. Note that distances must be calculated taking into account the periodic boundary conditions of the simulation box. All the distance matrices were computed using the Python package MDAnalysis[48]. For each of the statistically independent snapshots of the system, we evaluated the contact matrix. The final matrix, $H$, is given by the sum of the contact matrices for each snapshot.

In a similar way to what was done for the polymer contact matrix, we evaluated the contacts between the tracers (t particles) and the polymer. For each snapshot, we evaluated the contact matrix between all the tracers and all the particles that compose the polymer. We then considered as a contact any mutual distance shorter than $t$. The choice of $t$ is somewhat arbitrary. We set this value to $t = 2\sigma$, which seems a natural choice for it is the diameter of each particle, and also a distance at which the LJ interactions are practically zero. The tracer traffic at a monomer is then obtained by summing the contacts with all the tracers through all the snapshots of the simulation.

**Volume exclusion effects.** If there are many binding sites on the polymer (high $\phi$), its configuration in three dimensions will be compact, with the binding sites buried deep inside the globular part of the polymer. The more this is true, the more we expect the tracers to be excluded from contacting the binding sites that are in the globular core. Instead, we expect the tracers to bind more to the outer shell of the globule, which is more enriched with non-binding sites (i.e., particles of type p). To test these intuitive ideas and quantify volume exclusion effects on the diffusion of the tracers, we used several metrics.

First, we calculated the tracer coverage, defined as the percentage of monomers that are visited by the tracers during the simulation time frames. If some monomers are never contacted by the tracers because of volume exclusion, the coverage will be less than 100%.

Next, we calculated the Pearson correlation coefficient between the tracer traffic and the binder occupancy (calculated the same way as the tracer traffic). If the tracers are excluded from the polymer because of the binders, we expect that the two profiles will be anti-correlated.

**Biological data from GM12878.** The Hi-C data were downloaded from GEO (Rao et al.[30,31], accession ID GSE63525). Loop positions were obtained from the file `GSE63525_GM12878_primary+replicate_Arrowhead_domain-list.txt.gz`. The locations of the HOTs produced by Foley and Sidow[30] were downloaded from http://mendel.stanford.edu/sidowlab/downloads/hot/analysis/. We used the file `peaks_GM12878.fasta` and obtained the positions of the domains from the headers. The overlap between the two was computed with custom R scripts using the package GenomicRanges[49].

To compute the rates of contact decay, the raw Hi-C reads were normalized in 50 kb windows using the TADbit pipeline[50] with default parameters. The contact decays were estimated using a simple linear regression between the logarithm of the linear separation between the windows and the logarithm of the normalized Hi-C signal. For each window, only the closest 20 windows on each side were used for the regression, as the signal in windows beyond the 20th was typically too noisy. Also, the self contacts in the window were discarded for being overly influential on the regression parameters. The slope of the regression line served as estimate for the rate of contact decay.

**Code availability.** The code that was used to obtain and analyze the simulation data is available at https://github.com/rcortini/sbs_tracers (Zenodo https://doi.org/10.5281/zenodo.1201081).

**Data availability.** Data available on request from the authors.

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

## Acknowledgements

R.C. is grateful to Jean-Marc Victor, the members of the CNRS GDR 3536 (ADN), and Francesco Alessandro Massucci for useful discussions. The authors would like to thank Roni H.G. Wright for help with editing the manuscript, Egor Tiavlovsky for his help on an early version of the manuscript, and the CRG Scientific Information Technologies for helping with the simulation setup. We acknowledge the financial support of the Spanish Ministry of Economy and Competitiveness (Centro de Excelencia Severo Ochoa 2013–2017, SEV-2012–0208, Plan Nacional BFU2012-37168), of the CERCA Programme/Generalitat de Catalunya, of the European Research Council (ERC Synergy Grant 609989), and of the People Programme (Marie Curie Actions) of the European Union's Seventh Framework Programme (FP7/2007–2013) under REA grant agreement 608959.

## Author contributions

R.C. set up, ran, and analyzed all simulations. G.F. analyzed the data from GEO. R.C. and G.F. designed the study and wrote the manuscript.
