## [Peer Review File · Nature Communications]

Reviewers' comments:

Reviewer #1 (Remarks to the Author):

In this paper Cortini and Filion investigate the role of chromatin compaction in modulating the traffic of transcription factors. Using simulations based on a certain model of chromatin architecture, they make a number of interesting and novel predictions.

My over-all sentiment about this work is mixed. On the one hand, I believe that this field is in dire need of general theoretical models/frameworks that can help rein in the exploding zoo of experimental facts.

Currently, the field is dominated by experimental groups who publish highly specialized facts

in "Nature" or "Science", while

theoretical works that attempt to offer a synthesis and/or

a broader view of the elusive

"big picture" get sidelined. In that sense, the work by Cortini and

Filion is timely. Indeed, there is no "big picture" understanding

of how transcription is regulated by chromatin compaction. The general

argument the authors are trying to make (e.g. Fig. 7) has the potential to

advance the field. That is, if true, it may offer a significant advance towards the "big picture". The paper is written in good English.

However, in order to influence this experimentally dominated field, theoretical papers

must be of very high

quality, offering convincing arguments that will be hard to dismiss simply

because "this is yet another model". And that is where I think this work

is lacking significantly. I recommend a major revision.

MAJOR ISSUES.

1. How do we know that the regimes shown in e.g. Fig. 4 are relevant biologically? That is the range of model parameters explored by the authors has to be justified. How do we know that all of biology is not happening in just one corner of the diagram? The explanation for HOTs is consistent with the findings, but there can be a myriad of alternative explanations, so this one coincidence is nice, but not strongly convincing without a convincing argument for why the explored regimes are relevant.

2. I do believe that the argument the authors are trying to make in Fig. 7 is potentially strong: it is based on geometry that is hard to dismiss, and does not depend too much on details of the underlying model. However, it does depend critically on the geometry of the key objects to be just about right, see below. A justification along these lines is lacking.

2a. page 6. "In the first, the polymer forms globular domains that are too dense for tracers to enter. " Is this statement

supported by real physical dimensions of the tracers and chromatin structures relevant here?

2b. Likewise, on the next page "the polymer is never so dense as to be completely impermeable to tracers." Is that true? Playing Devil's advocate I would argue that the 30 nm fiber is packed densely enough for transcription factors to be unable to penetrate. We know now that the 30-nm fiber is unlikely to be the most prevalent structure, but this example shows the importance of thinking in terms of realistic numbers.

2c. Along the same lines, the authors state in Conclusions that "chromosome conformation effects apply to all the molecules in the nucleus with an affinity for chromatin. " Formally, this statement is just plain wrong: I am sure that e.g. water molecules are pretty much unaffected by the large scale re-arrangements of chromatin discussed by the authors. Again, all I am saying here is that the size matters: large molecules will be affected strongly, and small ones will not. What is this characteristic size? That is the key question left unanswered.

3. The "loop model" of chromatin compaction is just one model. There others, e.g. fractal globule. How robust is the key argument of Fig. 7 to the assumptions of the "loop model", e.g. in its "binders and switches" incarnation used by the authors? A more model-independent argument can probably be made here.

Other points:

4. "Transcription factors were represented by spherical particles referred to as tracers. Most importantly, tracers differ from transcription factors "

The statement is confusing. The second sentence contradicts the first: transcription factors are modeled as tracers, yet tracers are not transcription factors?

5. Please define "traffic" early on. Exactly what is it within your model? This is the key quantity of interest in this work.

6. The main message of Figure 2 is hard to grasp: there is simply too much information, and the data shows in too heterogeneous. The main quantity of interest is "transcription factor traffic". So why not define it in some meaningful way and show how it depends on the two key parameters of the model, epsilon and phi? That is just one "heat map" plot instead of the very dense and heterogeneous. Figure 4 is much clearer in that respect. Is " the average percentage of sites of the polymer that are visited by the tracers" = "traffic"?

7. Consider moving Fig. 7, or some generalized version of it, to the top of the "Results" section. This is the main result of the paper, well illustrated.

8. Why tracers "d" do not interact with "a" particles? How does this assumption affect the results? What is the ratio of a to b particles, and how it affects the results?

9. Please introduce Kullback-Leibler (KL) divergence and its physical meaning. Why is it a useful metric here? Again, I would suggest to define "traffic" mathematically, and focus on it.

10. page 9 "Hi-C contacts are locally proportional to $\frac{1}{s}$, where s is the linear separation between the loci and λ is the decay rate. Values of λ that are close to 0 indicate compact regions,"

I am assuming the authors mean "the probability of contacts"?

11. Also, the statement that values of the scaling exponent close to zero indicate compact regions is somewhat misleading.

Even for folded protein globules, which are as compact as they can be, s is never zero. I suggest choosing a better lower bound on s to indicate the compact regime.

Reviewer #2 (Remarks to the Author):

This paper discusses coarse-grained molecular dynamics simulations of a chromatin fibre interacting with a mixture of binders and tracers. The latter are used as a proxy/simple first model for proteins performing facilitated diffusion on chromatin. The main finding is that: (i) chromatin loops lead to colocalisation of binders and tracers for small chromatin+binders density; (ii) the positions of binders and tracers anticorrelate when the density of chromatin+binders is high. As the authors argue, some confirmation of the trends in their simple models are found by looking at the distributions of highly occupied targets (HOTs) genome-wide. I think the subject matter is interesting and this is a worthwhile investigation, which I enjoyed reading and which leads to some new results (although it should be put better in context in my view as I discuss later on). I am not though fully convinced that this work warrants publication in Nature Communications. The main issue with this is that the results come from a simple/toy model, and whilst interesting there are features of chromatin biophysics which are not included, and which may change the conclusions and their relevance to reality, in principle significantly.

Based on this I would find this work more appropriate for a slightly more specialised journal, such as Nucleic Acids Research or Scientific Reports. On the other hand if the authors find this concern to be misplaced, then I would be happy to review a revised version for Nature Communications. Below I list my concerns with the current version of the paper which led to this conclusion.

Novelty

In the current work the novelty lies in considering a mixture of specific binders and non-specific binders (tracers). Previous works considered both non-specific interactions (tracers) and specific interactions (binders) separately, and some of the results obtained in these models are at least superficially similar to those reported here (see below, Relation with previous literature). A key question to decide on whether this paper is suitable for the broad audience of Nature Communications is what advancement considering a *mixture* of tracers and binders constitutes, with respect to the previous works which considered each in isolation. To me, the difference seems to be interesting, but relatively minor, and the main conclusions are not that surprising if we know the previous works, by using some general polymer physics argument: we could expect looping to increase the density of chromatin and this should stimulate tracer binding, suitable to there being enough space (excluded volume effect). This is the reason why I am wondering whether a more specialised journal is more suitable. However I would be happy to know the authors' rebuttal on this main point.

Relation with previous literature

The authors correctly note that their model is a variation of the strings-and-binders model. However there are further results on related models which I think need to be included in the discussion. For instance, there seems to me to be some relation between the formation of clusters of tracers and the bridging-induced-attraction, which sets in for both specific and non-specific interactions, see

C. A. Brackley et al., Nonspecific bridging-induced attraction drives clustering of DNA-binding proteins and genome organization, *Proc. Natl. Acad. Sci. USA* 110, E3605 (2013); C. A. Brackley et al., Simulated binding of transcription factors to active and inactive regions folds human chromosomes into loops, rosettes and topological domains, *Nucleic Acids Res.* 44, 3503 (2016). These papers provide a feedback mechanism through which chromatin-binding proteins spontaneously cluster, and through which chromatin loops and clusters of loops (relevant for the structures studied here) are formed.

Still regarding the relation with other models, it seems to me that the relation to cohesin and loop extrusion (Refs. 35,36) is not discussed appropriately. In Refs. 35, 36, cohesin dimers are modelled as mobile springs which can move along the chromatin fibre. Here these are not included, hence I do not think the authors can draw too many conclusions about the importance of cohesins. Authors say that this work suggests a model where cohesin binding induces the formation of a chromatin loop, which increases the local concentration of transcription factors, which nucleates the formation of an aggregate stabilized by protein-protein interactions. But the looping in their model is due to the binders themselves, and the aggregate I would say forms through the feedback mechanism above. Hence I do not think these results can be really used to explain previous observations of factors colocalising with cohesins, and I recommend these comments should be reworded.

Model-related questions

A substantial simplification is that tracers are spherical and can make any number of contacts with chromatin (like in the strings-and-binders model). This may be appropriate for some but not all chromatin-binding proteins and this point should be discussed. I would imagine a tracer as a particle which minimally interferes with folding of chromatin, whereas multiple interactions as the authors note lead to local chromatin folding. Did the authors also try to consider tracers which can only make one contact with the chromatin fibre?

Many results in this work depend on the local density of chromatin and proteins. There are some new data on chromatin density and it would be nice to compare to those of the structures

simulated here, see

H. D. Ou et al, ChromEMT: Visualizing 3D chromatin structure and compaction in interphase and mitotic cells, Science 357 eaag0025 (2017). How do the local densities achieved here for chromatin compare with those inferred from the ChromEMT study?

Regarding now global, rather than local, density, the authors use a relatively dilute situation for chromatin of less than 1% in volume fraction (calculated with 1000 beads in 50 cube domain). But in reality the chromatin is more confined, and this crowding could potentially affect the traffic, at least quantitatively. I think this point deserves mention.

I do not understand why authors use Monte-Carlo for tracers which have no interactions (except for excluded volume) with the chromatin fibre. Molecular dynamics simulations with a purely repulsive Weeks-Chandler-Anderson potential should be perfectly OK to draw the same conclusions. In any case, I cannot see why including an interaction between tracer and chromatin helps: surely, one would expect that the molecular dynamics simulations with an attraction should be if anything slower to equilibrate with respect to those without interaction. Is the point that this does not matter in this case as kinetic effects are realistic? The current presentation is a bit confusing in terms of the relevance of equilibration and the distinction between Monte-Carlo versus molecular dynamics simulations.

In the Methods section the authors say that their diffusion coefficient is about $1 \text{ cm}^2/\text{s}$ for free spheres. But by using Stokes law with water-like viscosity and a bead size of 15 nm I get just above $30 \text{ micron}^2/\text{s}$, which is significantly less. Is there a typo here, or am I missing something?

Can the authors make a more quantitative comparison with the results of Ref. 20 for the facilitated diffusion of proteins on chromatin?

All these model-related questions are pretty minor, and will not require major modifications of the paper. The main issue regarding suitability is the novelty one, and I recommend that the authors deal with this in detail if they choose to resubmit to Nature Communications.

Reviewer #3 (Remarks to the Author):

Summary:

The manuscript of Cortini and Filion investigates through molecular dynamics and Monte Carlo simulations the consequences of unspecific interactions between transcription factors (TFs, tracers in the paper) and folded chromatin on TF occupancy. From the simulations, they could observe two major regimes. For low chromatin compaction (few binders), the TFs do not affect the 3D chromatin conformation and the profile of TF occupancies is driven by the number of contacts of the regions. On the contrary, for high compaction level, the polymer density prevents the binding of the TFs by volume exclusion and the number of contacts does not predict the probability of binding anymore. To support their results, the authors analyzed the genome-wide localization of Highly Occupied Targets regions (HOTs) that, according to the model, should be found close to chromatin loops.

Comments:

The study of how the chromatin architecture impacts different biological processes and in particular gene regulation has recently attract a lot of attention. The manuscript of Cortini and Filion definitely highlights an interesting connection between the 3D structure of chromatin and TF occupancy. The approach is sound however I have few concerns about the interpretation of the results. Thus, I would suggest some clarifications before publication.

Major comments:

- The authors claim that the occupancy profile observed at the regime of high polymer compaction

is explained by two factors: first, there is a core of polymer domains that become less accessible; second, anchor sites become crowded. It is not clear to what extent the first factor plays a role. For high compaction and low affinity 75% of the polymer is occupied at some moment by tracers. Can this just be explained by the competition between binders and tracers for anchor sites? Would longer simulation reach 100% coverage?

- In MD simulations, the polymer moves as opposed to the fix polymer structure used in MC simulations. Could the lack of correlation between the occupancy profiles be a consequence of using a fix structure vs a dynamic one? Moreover, it was not clear to me why simulating tracers without affinity for the polymer confirmed the effect of volume exclusion.

- Are the concentrations of binders and tracers used in the simulations realistic in normal physiological conditions?

Minor issues:

- I find that the nomenclature of binders and tracers makes the reading confusing since tracers are also binders. I would recommend to change it. Tracers and linkers, perhaps? The letters (a,b,c,d) chosen to denote the different players (as well as C and R) is not very helpful as they are unrelated to their names.

- To clarify this statement: "Second, the off-rate is reduced because the local concentration of the polymer increases, thereby favouring binding" to clarify the authors could add that this is caused by the attractive component of the potential.

- Typo: "We measured the correspondence between C and C"

- Typo or missing figure: "see Methods and figure 10"

We would first like to thank the reviewers for taking the time to review our manuscript and for their valuable comments. We aimed to be equally thorough and rigorous in our answers as they were in their reviews.

We named the points of the referee by RxPy, meaning "Reviewer number x, point number y". This way we could cross-reference easily the points, and give synthetic answers. Our answers to the points raised are written in green italics, below each point.

Reviewer #1 (Remarks to the Author):

In this paper Cortini and Fillion investigate the role of chromatin compaction in modulating the traffic of transcription factors. Using simulations based on a certain model of chromatin architecture, they make a number of interesting and novel predictions.

My overall sentiment about this work is mixed. On the one hand, I believe that this field is in dire need of general theoretical models/frameworks that can help rein in the exploding zoo of experimental facts. Currently, the field is dominated by experimental groups who publish highly specialized facts in "Nature" or "Science", while theoretical works that attempt to offer a synthesis and/or a broader view of the elusive "big picture" get sidelined. In that sense, the work by Cortini and Fillion is timely. Indeed, there is no "big picture" understanding of how transcription is regulated by chromatin compaction. The general argument the authors are trying to make (e.g. Fig. 7) has the potential to advance the field. That is, if true, it may offer a significant advance towards the "big picture". The paper is written in good English.

However, in order to influence this experimentally dominated field, theoretical papers must be of very high quality, offering convincing arguments that will be hard to dismiss simply because "this is yet another model". And that is where I think this work is lacking significantly. I recommend a major revision.

MAJOR ISSUES.

R1P1. How do we know that the regimes shown in e.g. Fig. 4 are relevant biologically? That is the range of model parameters deployed by the authors has to be justified. How do we know that all of biology is not happening in just one corner of the diagram? The explanation for HOTS is consistent with the findings, but there can be a myriad of alternative explanations, so this one coincidence is nice, but not strongly convincing without a convincing argument for why the explored regimes are relevant.

The varying parameters are the compaction of the polymer (ϕ) and the nonspecific affinity of the tracers for the polymer (ϵ).

Regarding ϕ , the visual aspect of experimental Hi-C maps is sufficient to convince oneself that chromatin can be in an "open" state where site-specific long-distance loops are visible, or in a "compacted" state where such loops do not occur. This is particularly visible for the active versus inactive X chromosomes in mouse (Giorgetti et al., Nature, 2016, PMID:

27437574), and in active versus inactive chromatin in *Drosophila* (Boettiger et al., *Nature*, 2016, PMID: 26760202). Recent estimates based on electron microscopy suggest that the density of chromatin varies by a factor 4 in the same nucleus (Ou et al., *Science* 2017, PMID: 28751582), which is consistent with previous estimates from Raman spectroscopy (Pliss et al., *Biophysical Journal* 2010, PMID: 21081098). By calculating the volume that the polymer occupies divided by the volume of the sphere of its radius of gyration (added in Supplementary Note 2), we can estimate the density in our simulations. Matching the estimated density with the results of Ou et al. suggests that in human cells, ϕ ranges in 0.1-0.4. This information was added to the Results and Methods sections.

Regarding ϵ , estimates of the nonspecific affinity of nuclear proteins for chromatin in vivo are scarce in the scientific literature, but experimental measurements in mammalian cells suggest that TetR spends approximately the same amount of time unbound to chromatin and associated to nonspecific sites (Normanno et al., *Nature Communications*, 2015, PMID: 26151127). In our setup, this corresponds to a value of ϵ equal to 2.0 kT (see Supplementary Figure 3C). The minimum value of ϵ is less relevant because it describes molecules without affinity for the polymer, which is unrelated to our study. In conclusion, it is reasonable to assume that ϵ can reach at least 2.0 kT in biological systems. This information was added to the Results and Methods sections.

R1P2. I do believe that the argument the authors are trying to make in Fig. 7 is potentially strong: it is based on geometry that is hard to dismiss, and does not depend too much on details of the underlying model. However, it does depend critically on the geometry of the key objects to be just about right, see below. A justification along these lines is lacking.

R1P2a. page 6. "In the first, the polymer forms globular domains that are too dense for tracers to enter." Is this statement supported by real physical dimensions of the tracers and chromatin structures relevant here?

*The point that the Referee raises here is interesting and important. We added a discussion on the effect of the size of the tracers in Supplementary Note 5. Our results show that the main predicted effects (the principles of transcription factor traffic) are relevant when the size of the transcription factors is comparable to the size of the chromatin structures (i.e. the DNA, the nucleosome, the chromatin loops, etc.). Since the size of the loops is closely related to the polymer compaction, probed by ϕ (see, e.g., Supplementary Figure 1C-D), we chose in the beginning of our simulation setup that we would express the results as a function of the polymer compaction and not of the tracer diameter. We also show that the correlations between the polymer structure and the tracer traffic can be non-negligible even when the size of the tracers is 5-10 times smaller than the size of the monomers, which also shows that our results are robust. From the biological point of view, the sizes that we deal with here are relevant, since the transcription factor diameter can go from about 5 nm to about 20 nm (see, e.g. Maeshima et al., *Journal of Physics: Condensed Matter*, 2015, PMID: 25563431).*

R1P2b. Likewise, on the next page "the polymer is never so dense as to be completely impermeable to tracers." Is that true? Playing Devil's advocate I would argue that the 30 nm fiber is packed densely enough for transcription factors to be unable to penetrate. We know now that the 30-nm fiber is unlikely to be the most prevalent structure, but this example shows

the importance of thinking in terms of realistic numbers.

The statement referred only to the results of the simulations and did not have the value of a general truth. Some structures are probably impermeable, but as pointed out by the Reviewer, the question is whether they occur in vivo. Experimental evidence suggests that proteins with an affinity for chromatin are able to penetrate the most densely packed compartments in cell nuclei (Bancaud et al, EMBO Journal, 2009, PMID: 19927119). In our simulations, the highest values of phi results in a three-dimensional structure of the polymer that does not allow tracers with low affinity to penetrate its core. However, with the highest values of the affinity that we probed, no region of the polymer is impenetrable from the tracers, no matter how high the value of phi is. This is totally in line with the experimental results, and stands in favour of the plausibility of our model.

To clarify that we do not mean that tight polymer structures are impossible, we rephrased this sentence as "In fact most of the monomers are accessible at some value of the non-specific affinity, so for the highest values of the compaction that we probed, the polymer is never so dense as to be completely impermeable to tracers. We surmise that higher values of the compaction would result in a completely impenetrable polymer core".

R1P2c. Along the same lines, the authors state in Conclusions that "chromosome conformation effects apply to all the molecules in the nucleus with an affinity for chromatin. " Formally, this statement is just plain wrong: I am sure that e.g. water molecules are pretty much unaffected by the large scale re-arrangements of chromatin discussed by the authors. Again, all I am saying here is that the size matters: large molecules will be affected strongly, and small ones will not. What is this characteristic size? That is the key question left unanswered.

Thank you for pointing out this inaccuracy. We changed now the text to "The key point is that chromosome conformation effects apply to all the molecules in the nucleus with an affinity for chromatin, provided that the size of the particles is of the order of the size of the pores of the chromosome folds (see Supplementary Note 5)."

R1P3. The "loop model" of chromatin compaction is just one model. There others, e.g. fractal globule. How robust is the key argument of Fig. to the assumptions of the "loop model", e.g. in its "binders and switches" incarnation used by the authors? A more model-independent argument can probably be made here.

This is an important point of discussion. The claim that transcription factors must spend more time at the more connected sites of the genome is very robust to model assumptions. The best demonstration is that this results still holds for toy models of random walks on networks. This amounts to performing an intermittent 1D / 3D search directly on the contact matrix (see for instance Avcu and Molina, bioRxiv 2016, <https://doi.org/10.1101/050146>), which does not even use polymer physics. The molecular dynamics simulations are necessary to capture volume exclusion effects that appear when the polymer is crowded or compacted. So the claim that transcription factors are less likely to enter a compact chromatin domain depends on the assumptions of the model.

The first claim is by far the most relevant for this work and we are confident that it depends very little on the assumptions of the model. As for the second claim, it is so intuitive that it seems difficult to think of realistic models where the opposite would occur.

Just to be clear, in this work we have adopted a phenomenological viewpoint. This means that we use the models to represent 'generic loops' in 'generic polymers'. At no point do we claim that this model represents actual molecular mechanisms in vivo. Since the main conclusion is validated with much fewer assumptions (e.g., random walks on networks), we are confident that the representation we chose describes the high-level behavior of the system.

Other points:

R1P4. "Transcription factors were represented by spherical particles referred to as tracers. Most importantly, tracers differ from transcription factors " The statement is confusing. The second sentence contradicts the first: transcription factors are modeled as tracers, yet tracers are not transcription factors?

We clarified this point. The text now reads "We simulated spherical particles, referred to as tracers, which are a model for transcription factors except for one important difference: tracers have no specific affinity for any particular site on the polymer, contrary to transcription factors which have a specific target site."

R1P5. Please define "traffic" early on. Exactly what is it within your model? This is the key quantity of interest in this work.

*We now defined "traffic" as per the suggestion of next point and introduce the definition when describing Figure 2. We also avoid mentioning **C** and **R**, for clarity. We now consistently speak only of "polymer contacts" and "tracer traffic".*

R1P6. The main message of Figure 2 is hard to grasp: there is simply too much information, and the data shows in too heterogeneous. The main quantity of interest is "transcription factor traffic". So why not define it in some meaningful way and show how it depends on the two key parameters of the model, epsilon and phi? That is just one "heat map" plot instead of the very dense and heterogeneous. Figure 4 is much clearer in that respect. Is "the average percentage of sites of the polymer that are visited by the tracers" = "traffic"?

We have redesigned Figure 2 along those guidelines and those of the next point. Figure 2 now shows the "traffic" instead of the "occupancy".

R1P7. Consider moving Fig. 7, or some generalized version of it, to the top of the "Results" section. This is the main result of the paper, well illustrated.

The information that was previously in Figure 7 is now a panel of Figure 2. The text has been updated accordingly.

R1P8. Why tracers "d" do not interact with "a" particles? How does this assumption affect the

results? What is the ratio of a to b particles, and how it affects the results?

In fact, tracers do interact with “a” particles. We have clarified this in the Methods section. The number of “b” particles is equal to $N\phi$, where ϕ is our control parameter. The ratio of “a” to “b” particles is therefore $(1-\phi)/\phi$. We illustrate the results as a function of ϕ .

NB: we have changed the letters designating particles as per recommendation of Reviewer 3, but in this answer we still used the previous nomenclature.

R1P9. Please introduce Kullback-Leibler (KL) divergence and its physical meaning. Why is it a useful metric here? Again, I would suggest to define "traffic" mathematically, and focus on it.

The KL divergence quantifies the information lost by using the contact profile as a proxy for the traffic (it is a standard way to compare distributions). Since the contact profile and the traffic can be viewed as distributions or as variables, we compare them with the KL divergence and with the Pearson correlation coefficient. There are cases in which using the correlation coefficient alone would lead to a misleading interpretation of the results (see below our results concerning the new simulations with monovalent tracers).

We clarified the meaning of the Kullback-Leibler at the beginning of the “Results” section. We expanded the discussion on its interpretation and importance in the second paragraph of the Results section.

R1P10. page 9 "Hi-C contacts are locally proportional to $\frac{1}{s}$, where s is the linear separation between the loci and λ is the decay rate. Values of λ that are close to 0 indicate compact regions," I am assuming the authors mean "the probability of contacts"?

Yes, we clarified the text and incorporated the referee’s suggestion.

R1P11. Also, the statement that values of the scaling exponent close to zero indicate compact regions is somewhat misleading. Even for folded protein globules, which are as compact as they can be, s is never zero. I suggest choosing a better lower bound on s to indicate the compact regime.

Thank you for pointing out this phrasing issue. The exponent being negative, we just meant “smaller in absolute value”. We made the text clearer by defining α as a positive value so that the scaling exponent is minus α . The expression “smaller values of α ” is now less ambiguous.

Reviewer #2 (Remarks to the Author):

This paper discusses coarse-grained molecular dynamics simulations of a chromatin fibre interacting with a mixture of binders and tracers. The latter are used as a proxy/simple first

model for proteins performing facilitated diffusion on chromatin. The main finding is that: (i) chromatin loops lead to colocalisation of binders and tracers for small chromatin+binders density; (ii) the positions of binders and tracers anticorrelate when the density of chromatin+binders is high. As the authors argue, some confirmation of the trends in their simple models are found by looking at the distributions of highly occupied targets (HOTs) genome-wide. I think the subject matter is interesting and this is a worthwhile investigation, which I enjoyed reading and which leads to some new results (although it should be put better in context in my view as I discuss later on). I am not though fully convinced that this work warrants publication in Nature Communications. The main issue with this is that the results come from a simple/toy model, and whilst interesting there are features of chromatin biophysics which are not included, and which may change the conclusions and their relevance to reality, in principle significantly. Based on this I would find this work more appropriate for a slightly more specialised journal, such as Nucleic Acids Research or Scientific Reports. On the other hand if the authors find this concern to be misplaced, then I would be happy to review a revised version for Nature Communications. Below I list my concerns with the current version of the paper which led to this conclusion.

Novelty

R2P1. In the current work the novelty lies in considering a mixture of specific binders and non-specific binders (tracers). Previous works considered both non-specific interactions (tracers) and specific interactions (binders) separately, and some of the results obtained in these models are at least superficially similar to those reported here (see below, Relation with previous literature). A key question to decide on whether this paper is suitable for the broad audience of Nature Communications is what advancement considering a *mixture* of tracers and binders constitutes, with respect to the previous works which considered each in isolation. To me, the difference seems to be interesting, but relatively minor, and the main conclusions are not that surprising if we know the previous works, by using some general polymer physics argument: we could expect looping to increase the density of chromatin and this should stimulate tracer binding, suitable to there being enough space (excluded volume effect). This is the reason why I am wondering whether a more specialised journal is more suitable. However I would be happy to know the authors' rebuttal on this main point.

This work is the first explicit demonstration that the geometry of the chromatin polymer can explain biologically relevant patterns of transcription factor binding. To recapitulate, we discover that Highly Occupied Targets are enriched at the anchors of chromatin loops and we propose an explanation based on simple laws of physics. Meanwhile we formulate the main principles of a theory explaining how transcription factors can spend more time searching their target in some regions than in others. This boils down to genome geometry, which was never mentioned in a biological context.

The novelty of this work is not the assumptions, but the explanations. The models are purposely simple in order to be as general as possible. One of the main strengths of the manuscript is that it relies only on straightforward assumptions and undisputed effects. The outcome may not surprise an expert of polymer physics, but the conclusions are decidedly in the realm of biology, where they are far from trivial. The notion that the geometry of the genome can influence the distribution of transcription factors is extremely important to

understand the heterogeneity of the nucleus. And yet, this message is entirely absent from the biological literature.

More than anything else, the novelty of this work resides in the combination of ideas. Bits and pieces of the main arguments were published in different articles, but the concepts are reunited in a consistent synthesis for the first time. This makes the article a landmark in this regard. For instance, the transition from 'open' to 'closed' regimes described in the text was not fully investigated in this context, Highly Occupied Targets were not linked to chromatin loops, nor were they considered from the point of view of polymer physics. More generally, the mechanistic consequences of enhancer-promoter looping have been mostly disregarded so far.

Our ambition here is to speak to biologists using a language from polymer physics. The question is not "how many" people read Nature Communication but "who" reads it. In the recent years, Nature Communication has imposed itself as the reference journal for interdisciplinary research, which is in perfect line with our declared goal.

We particularly appreciate the respectful tone of the Reviewer on this question. One could envision other ways to communicate our results in an engaging way to the target audience, and we are open to further suggestions on how to make the article interesting for both physicists and biologists.

Relation with previous literature

R2P1 (continued). The authors correctly note that their model is a variation of the strings-and-binders model. However there are further results on related models which I think need to be included in the discussion. For instance, there seems to me to be some relation between the formation of clusters of tracers and the bridging-induced-attraction, which sets in for both specific and non-specific interactions, see C. A. Brackley et al., Nonspecific bridging-induced attraction drives clustering of DNA-binding proteins and genome organization, Proc. Natl. Acad. Sci. USA 110, E3605 (2013); C. A. Brackley et al., Simulated binding of transcription factors to active and inactive regions folds human chromosomes into loops, rosettes and topological domains, Nucleic Acids Res. 44, 3503 (2016). These papers provide a feedback mechanism through which chromatin-binding proteins spontaneously cluster, and through which chromatin loops and clusters of loops (relevant for the structures studied here) are formed.

We now cite the two papers that the referee suggests.

We agree that our model is similar to that proposed by Marenduzzo and collaborators, but we pursue a different objective. As highlighted above, our ambition is to give quantitative insight into the relationship between a given polymer conformation and the distribution of transcription factors. Here, the structure of the polymer is given, and our focus is not on how that structure is formed in the first place. The two studies give further support to our claims, by stressing even more that the relationship between the large-scale conformation of the polymer and the distribution of tracers have to be intimately related. Moreover, our present work gives a quantitative account of the volume exclusion effect, which was not treated in the

papers mentioned above.

R2P2. Still regarding the relation with other models, it seems to me that the relation to cohesin and loop extrusion (Refs. 35,36) is not discussed appropriately. In Refs. 35, 36, cohesin dimers are modelled as mobile springs which can move along the chromatin fibre. Here these are not included, hence I do not think the authors can draw too many conclusions about the importance of cohesins. Authors say that this work suggests a model where cohesin binding induces the formation of a chromatin loop, which increases the local concentration of transcription factors, which nucleates the formation of an aggregate stabilized by protein-protein interactions. But the looping in their model is due to the binders themselves, and the aggregate I would say forms through the feedback mechanism above. Hence I do not think these results can be really used to explain previous observations of factors colocalising with cohesins, and I recommend these comments should be reworded.

Thank you for pointing out the lack of clarity. This passage does not refer to the cohesin loop extrusion model of references 35 and 36 (now references 36 and 37), but to the experimental observations that cohesin is necessary for the formation of loops (Rao et al., Cell 2017, reference 43) and that it is the protein most frequently present in Highly Occupied Targets (Yan et al., Cell 2013, reference 42). This latter fact is somewhat overlooked, explaining that most readers will assume that we talk about the more popular but still speculative loop extrusion model.

With these expectations in mind, we wrote this passage to be consistent with the current view on loop extrusion. The cohesin rings are assumed to move along the chromosome until they meet CTCF. In this model, loops are dynamic but they still spend long periods of time in a static state (which gives the genome its conformation). Experimentally, it was found that cohesin characteristic binding time is very long (Hansen et al., eLife 2017, reference 44). We reiterate that what creates the loops or how they form has little relevance for our conclusions.

We fully agree with the Reviewer that we cannot draw any conclusion about the importance of cohesin from this work. But we do not need to: the necessity of cohesin in the formation of at least some chromatin loops is an undisputed experimental fact (Rao et al., Cell 2017, reference 43). We surmise that this is the point that caused confusion and we have rephrased the text to clarify this.

Model-related questions

R2P3. A substantial simplification is that tracers are spherical and can make any number of contacts with chromatin (like in the strings-and-binders model). This may be appropriate for some but not all chromatin-binding proteins and this point should be discussed. I would imagine a tracer as a particle which minimally interferes with folding of chromatin, whereas multiple interactions as the authors note lead to local chromatin folding. Did the authors also try to consider tracers which can only make one contact with the chromatin fibre?

We thank the Reviewer for highlighting this important point. We performed another large-scale simulation round with monovalent tracers. We included the description of the simulation setup and results in the Supplementary Note 4. With a few minor differences, our

results hold in their essence even in the case of monovalent tracers. We also included a sentence in the Results section of the Main Text to underline that our results are model-independent also from this point of view.

R2P4. Many results in this work depend on the local density of chromatin and proteins. There are some new data on chromatin density and it would be nice to compare to those of the structures simulated here, see H. D. Ou et al, ChromEMT: Visualizing 3D chromatin structure and compaction in interphase and mitotic cells, Science 357 eaag0025 (2017). How do the local densities achieved here for chromatin compare with those inferred from the ChromEMT study?

The estimates of the local density provided in the work of Ou et al. are indeed very useful because they provide us with means of evaluating which are the values of ϕ that are physiologically relevant. We assessed the local polymer density by computing the radius of gyration of the polymer and by evaluating the volume that the monomers occupy in a sphere that has the radius corresponding to the polymer's radius of gyration. As a result of this estimate, we can say that the results of ChromEMT correspond to the case of ϕ going from 10% to 40%. See also the response to R1P1.

R2P5. Regarding now global, rather than local, density, the authors use a relatively dilute situation for chromatin of less than 1% in volume fraction (calculated with 1000 beads in 50 cube domain). But in reality the chromatin is more confined, and this crowding could potentially affect the traffic, at least quantitatively. I think this point deserves mention.

We thank the Reviewer for this very interesting comment, which prompted us to perform a new simulation set. The results of our new simulations are discussed in Supplementary Note 6, and are performed using the same conditions as before, but with the presence of additional crowding factors, which have the only effect of excluding the volume for the tracers and the polymer. The result, as we show, is that the effect of macromolecular crowding is to increase the value of the tracer-polymer affinity (by increasing the effective concentration of the tracers). For the other part, the results of our simulations hold even in the case of crowding up to 30% of the volume of the simulations.

R2P6. I do not understand why authors use Monte-Carlo for tracers which have no interactions (except for excluded volume) with the chromatin fibre. Molecular dynamics simulations with a purely repulsive Weeks-Chandler-Anderson potential should be perfectly OK to draw the same conclusions. In any case, I cannot see why including an interaction between tracer and chromatin helps: surely, one would expect that the molecular dynamics simulations with an attraction should be if anything slower to equilibrate with respect to those without interaction. Is the point that this does not matter in this case as kinetic effects are realistic? The current presentation is a bit confusing in terms of the relevance of equilibration and the distinction between Monte-Carlo versus molecular dynamics simulations.

We agree with the Referee that the presentation of our results was unclear. We therefore decided to remove the part about Monte Carlo simulations, and instead present the results of Molecular Dynamics simulations in the absence of interactions between the tracers and the polymer. To speed up convergence time, in these control simulations we kept the polymer

fixed and allow for the movement only of the tracers, which interact only via hard-core repulsion with the other particles in the system. The results are the same as in the previous version, and the manuscript is more consistent because we use Molecular Dynamics simulations throughout.

R2P7. In the Methods section the authors say that their diffusion coefficient is about $1 \text{ cm}^2/\text{s}$ for free spheres. But by using Stokes law with water-like viscosity and a bead size of 15 nm I get just above $30 \text{ micron}^2/\text{s}$, which is significantly less. Is there a typo here, or am I missing something?

The value of the viscosity in our simulations was not chosen to match that of water, but to make the tracers go fast enough to quickly equilibrate the system. See the answer above for a discussion on the difference between equilibrium and kinetic properties of the system.

R2P8. Can the authors make a more quantitative comparison with the results of Ref. 20 for the facilitated diffusion of proteins on chromatin?

We made a quantitative comparison to Ref. 20 in discussing the relationship between the values of epsilon and the amount of time that the tracers spend bound to the polymer (see Supplementary Figure 2C). From this comparison, we deduce that the value of epsilon is around 2 kT in their experimental system, which is roughly the value corresponding to a few hydrogen bonds forming between a protein and the chromatin. This is discussed in the new paragraph we added in the introduction to the Results section.

All these model-related questions are pretty minor, and will not require major modifications of the paper. The main issue regarding suitability is the novelty one, and I recommend that the authors deal with this in detail if they choose to resubmit to Nature Communications.

Reviewer #3 (Remarks to the Author):

Summary:

The manuscript of Cortini and Filion investigates through molecular dynamics and Monte Carlo simulations the consequences of unspecific interactions between transcription factors (TFs, tracers in the paper) and folded chromatin on TF occupancy. From the simulations, they could observe two major regimes. For low chromatin compaction (few binders), the TFs do not affect the 3D chromatin conformation and the profile of TF occupancies is driven by the number of contacts of the regions. On the contrary, for high compaction level, the polymer density prevents the binding of the TFs by volume exclusion and the number of contacts does not predict the probability of binding anymore. To support their results, the authors analyzed the genome-wide localization of Highly Occupied Targets regions (HOTs) that, according to the model, should be found close to chromatin loops.

Comments:

The study of how the chromatin architecture impacts different biological processes and in particular gene regulation has recently attract a lot of attention. The manuscript of Cortini and Fillon definitely highlights an interesting connection between the 3D structure of chromatin and TF occupancy. The approach is sound however I have few concerns about the interpretation of the results. Thus, I would suggest some clarifications before publication.

Major comments:

R3P1. The authors claim that the occupancy profile observed at the regime of high polymer compaction is explained by two factors: first, there is a core of polymer domains that become less accessible; second, anchor sites become crowded. It is not clear to what extend the first factor plays a role. For high compaction and low affinity 75% of the polymer is occupied at some moment by tracers. Can this just be explained by the competition between binders and tracers for anchor sites? Would longer simulation reach 100% coverage?

We have performed simulations on a 10-fold longer time scale and have observed that the coverage increases (see Supplementary Figure 6A). Based on theoretical arguments, we expect that coverage would reach 100% for infinite running times, but the results above suggest that the value reaches a plateau during the time frame set in our simulations and increase exceedingly slowly from there. We are thus confident that our results are representative of the equilibrium and do not of a transient state of the system.

We have added this information in Supplementary Note 7.

R3P2. In MD simulations, the polymer moves as opposed to the fix polymer structure used in MC simulations. Could the lack of correlation between the occupancy profiles be a consequence of using a fix structure vs a dynamic one? Moreover, it was not clear to me why simulating tracers without affinity for the polymer confirmed the effect of volume exclusion.

As for the answer to R2P6, we agree that the way our MC simulation results were presented was confusing. Therefore we decided to perform MD simulations without any tracer-polymer affinity. In these simulations the polymer is kept fixed, and we verified that the traffic on the polymer in this case is highly correlated to the one with the moving polymer. This is due to the fact that the polymer structure is very rigid due to the high strength of the interactions between anchors and binders. The results of this new simulation round are the same as in the previous version.

Tracers without affinity for the polymer interact with it only through volume exclusion. The fact that such tracers become depleted from loop anchors in compact configurations can thus only be attributed to volume exclusion. Since volume exclusion is sufficient to explain this effect, we do not need to introduce additional hypotheses to justify this phenomenon when tracers have an affinity for the polymer (in which case volume exclusion also takes place).

We clarified this by adding a final summary paragraph at the end of the section named “High polymer compaction excludes tracers”.

R3P3. Are the concentrations of binders and tracers used in the simulations realistic in normal physiological conditions?

This is a good point. The concentration of binders is not relevant per se because they are only used to create a certain amount of loops (this is the phenomenological viewpoint mentioned at R2P3). More relevant for the simulations is the number of loops and the associated compaction, which we discussed extensively R1P1.

Throughout the work, we used two concentrations of tracers to cover the possible cases. In our models, the monomers do not correspond to a concrete reference, but based on common knowledge on DNA and chromatin, they must lie in the range 0.2-2 kb of genome. This means that at the lower concentration, tracers are present at 1 molecule per 20-200 kb of genome, and 20 times as much at the higher concentration. In comparison, RNA polymerase II is present in ~300k copies per nucleus (Kimura et al., Molecular and Cellular Biology, 1999, PMID: 10409729), i.e. 1 molecule per 20 kb of genome. For typical transcription factors, the estimates range from less than 6k to more than 3000k copies per nucleus, depending on the molecule and the cell type (Biggin, Developmental Cell, 2011, PMID: 22014521), i.e. 1 molecule per 1000 kb to 1 molecule per 2 kb of genome. Even though this range is very wide, it shows that the concentrations of tracers in the simulations is realistic.

Minor issues:

R3P4. I find that the nomenclature of binders and tracers makes the reading confusing since tracers are also binders. I would recommend to change it. Tracers and linkers, perhaps? The letters (a,b,c,d) chosen to denote the different players (as well as C and R) is not very helpful as they are unrelated to their names.

We absolutely agree with this comment, but we have to stick to the nomenclature proposed by Barbieri et al. (reference 25) because we use the same model. Changing convention would make the work easier to understand for the readers who are not familiar with the model of Barbieri et al. but it would make it harder for the readers who are. It would also raise many

unnecessary questions regarding what assumptions we have changed. It is somewhat unfortunate that Barbieri et al. chose the word 'binder', but in their defense, it was the only species binding the polymer in their original model.

That said, if the opinion of the Editor is that clarity of the current manuscript is more important than consistency with the previous literature, we will be happy to oblige and replace 'binders' by 'linkers'.

Concerning the letters assigned to each particle, we agree with the Referee and we have changed the particle letters to p (polymer, was "a"), a (anchor, was "b"), b (binder, was "c") and t (tracer, was "d").

R3P5. To clarify this statement: "Second, the off-rate is reduced because the local concentration of the polymer increases, thereby favouring binding" to clarify the authors could add that this is caused by the attractive component of the potential.

Thank you for this suggestion. We have clarified the statement following those guidelines.

R3P6. Typo: "We measured the correspondence between C and C"

We deleted this sentence because we now always refer to "polymer contacts" and "tracer traffic", avoiding to use symbols that are not clear (see also R1P5, R1P6).

R3P7. Typo or missing figure: "see Methods and figure 10"

Yes, thank you for pointing this out, we deleted the sentence because it corresponded to an earlier version of the manuscript.

REVIEWERS' COMMENTS:

Reviewer #1 (Remarks to the Author):

The authors have addressed my concerns in a reasonable way.

Last comment: please make sure that your own statement (or its equivalent) "At no point do we claim that this model represents actual molecular mechanisms in vivo." ends up in the paper, preferably in "Conclusions".

Alexey Onufriev,
Virginia Tech

Reviewer #2 (Remarks to the Author):

In this revision the authors have done a good job in my view of addressing most of my technical comments on the model. I can also better see their general viewpoint in the reply, that this model is useful because it starts from simple models leading to a network of loops (without worrying about the mechanism leading to cluster formation), and show that once this network has formed, then proteins binding chromatin non-specifically (tracers) may focus on the base/anchors of the loops for suitable parameter values. It is interesting that this qualitatively explains the formation of patterns of HOTs at regions rich in HiC contacts and given the size of the biology and biophysics community which is interested in these topics, I agree this message can find a good audience in Nat. Comm.

I still have a few remaining (minor) problems with the discussion of these results, which I do not think currently conveys the message above (from the reply) clearly enough. These are optional, but I hope the authors will consider them as I think will improve readability.

Mechanism

On page 5 of the pdf the authors say that there is an effect on the on rate, and one on the off rate. Whilst these two statements are correct, I would say the simulated ChIP-seq is essentially simply counting the average number of non-specific interactions which the tracers can make (this is related to the off/on rate, rather than the rates separately). If the authors agree that this is the case, then it should be said more clearly/added when Figure 2 is introduced, as it leads to a simpler understanding of the effect (at least in my view).

Discussion and conclusions

The authors also state as a main result the fact that tracers nucleate clustering/compaction (5th paragraph onwards in the Discussion). I would reduce this claim, as this part was essentially shown before.

For instance, the authors mention that strings-and-binders and bridging-induced attraction refer to cases without loops but this is not really the case, e.g. in ref. 41 proteins have a combination of specific and non-specific interactions so the chromatin also forms a very similar network of loops (there is also other later work with similar assumption and results). Why not simply saying that here authors considered a mixture of proteins interacting non-specifically and proteins interacting specifically whereas previous models considered only one set of proteins with both interactions?

The subsequent part on cohesin and liquid phase separation is still a bit speculative and less convincing than the discussion of HOTs. It feels a bit tangential to the main message, hence I would recommend shrinking it down considerably if possible.

E.g., the view that the model predicts that cohesin KO leads to HOT disappearance/change is still unconvincing. For instance HOT formation requires a cluster of several loops and (despite the

authors' reply) it is unclear that cohesin is sufficient (or even necessary) for such loop clusters, as for instance super-enhancer clusters appear following cohesin removal (SSP Rao et al, Cell 171, 305 (2017)).

If this last part on cohesin/liquid droplet formation is shrunk it also stresses more the novelty of the HOT discussion, which I think is the really key contribution.

Reviewer #3 (Remarks to the Author):

The authors addressed satisfactory the issues I posed. I believe the manuscript has improved in clarity and the results are interesting and novel enough to be published in nature communications.

Reviewer #1 (Remarks to the Author):

The authors have addressed my concerns in a reasonable way.

Last comment: please make sure that your own statement (or its equivalent) "At no point do we claim that this model represents actual molecular mechanisms in vivo." ends up in the paper, preferably in "Conclusions".

Alexey Onufriev,
Virginia Tech

We thank the Reviewer. We included a statement in the first paragraph of the Discussion: "Despite the fact that our assumptions are reasonable, our model of chromosomes and transcription factors is simplistic, so we do not claim that it represents the real molecular detail at work in a cell nucleus. It only captures some general properties of the system."

We also added the following sentence at the end of the paragraph where we speculate about the role of cohesin proteins: "Once again, our model does not represent actual molecular mechanisms but only their essential properties regarding the diffusion of transcription factors".

Reviewer #2 (Remarks to the Author):

In this revision the authors have done a good job in my view of addressing most of my technical comments on the model. I can also better see their general viewpoint in the reply, that this model is useful because it starts from simple models leading to a network of loops (without worrying about the mechanism leading to cluster formation), and show that once this network has formed, then proteins binding chromatin non-specifically (tracers) may focus on the base/anchors of the loops for suitable parameter values. It is interesting that this qualitatively explains the formation of patterns of HOTs at regions rich in HiC contacts and given the size of the biology and biophysics community which is interested in these topics, I agree this message can find a good audience in Nat. Comm.

We thank the Referee.

I still have a few remaining (minor) problems with the discussion of these results, which I do not think currently conveys the message above (from the reply) clearly enough. These are optional, but I hope the authors will consider them as I think will improve readability.

Mechanism

On page 5 of the pdf the authors say that there is an effect on the on rate, and one on the off rate. Whilst these two statements are correct, I would say the simulated ChIP-seq is essentially simply counting the average number of non-specific interactions which the tracers can make (this is related to the off/on rate, rather than the rates separately). If the authors

agree that this is the case, then it should be said more clearly/added when Figure 2 is introduced, as it leads to a simpler understanding of the effect (at least in my view).

We agree with the Referee. We added a sentence at the end of the corresponding paragraph: “The overall effect is an increase of the ratio between on-rate and off-rate, *i.e.* a net increase in traffic.”

Discussion and conclusions

The authors also state as a main result the fact that tracers nucleate clustering/compaction (5th paragraph onwards in the Discussion). I would reduce this claim, as this part was essentially shown before.

For instance, the authors mention that strings-and-binders and bridging-induced attraction refer to cases without loops but this is not really the case, e.g. in ref. 41 proteins have a combination of specific and non-specific interactions so the chromatin also forms a very similar network of loops (there is also other later work with similar assumption and results). Why not simply saying that here authors considered a mixture of proteins interacting non-specifically and proteins interacting specifically whereas previous models considered only one set of proteins with both interactions?

We modified the paragraph, which now reads:

“We showed, as mentioned in the Results section, that when the number of tracers and their affinity exceed a critical value, the tracers themselves create intra-polymer contacts (Supplementary Note 2). This effect was predicted by the strings and binders switch model of Barbieri et al. (2012) and the bridging-induced attraction mechanism (Brackley et al. 2013, Brackley et al. 2016), in which loops are created in a non-specific fashion. Here, we show that it is also valid when the polymer has specific loops in its structure, due to the fact that we separated the role of specific and non-specific bridging particles. This indicates that a high concentration of transcription factors may by itself induce the chromatin fiber to collapse locally, a result which suggests that the many transcription factors present in HOTs may stabilize long-range contacts. More work is necessary to clarify the effect of heterogeneous binding profiles of the transcription factors in mediating and stabilizing chromatin loops.”

The subsequent part on cohesin and liquid phase separation is still a bit speculative and less convincing than the discussion of HOTs. It feels a bit tangential to the main message, hence I would recommend shrinking it down considerably if possible.

E.g., the view that the model predicts that cohesin KO leads to HOT disappearance/change is still unconvincing. For instance HOT formation requires a cluster of several loops and (despite the authors' reply) it is unclear that cohesin is sufficient (or even necessary) for such loop clusters, as for instance super-enhancer clusters appear following cohesin removal (SSP Rao et al, Cell 171, 305 (2017)).

If this last part on cohesin/liquid droplet formation is shrunk it also stresses more the novelty of the HOT discussion, which I think is the really key contribution.

We see the referee's point. We agree that some of this discussion is more an interpretation than a direct consequence of the work presented in the manuscript. That said, we think that providing testable predictions is important and gives the experimental community the

opportunity to directly verify that our predictions are correct. We have decided to remove the prediction that HOTs should disappear upon cohesin removal (for the reasons given by the Reviewer), but we have kept the other two predictions.

We do not think that speculation should be removed completely, but we have shortened the end of the Discussion in order to conclude the paper more rapidly and, as suggested by the Reviewer, give more emphasis to the core of our conclusions.

In any case, we will follow the recommendations of the Editor in this regard.

Reviewer #3 (Remarks to the Author):

The authors addressed satisfactory the issues I posed. I believe the manuscript has improved in clarity and the results are interesting and novel enough to be published in nature communications.

We thank the Referee.